# Evaluation of biomechanically corrected intraocular pressure using Corvis ST and comparison of the Corvis ST, noncontact tonometer, and Goldmann applanation tonometer in patients with glaucoma

**Yoshitaka Nakao⍟\*, Yoshiaki Kiuchi, Hideaki Okumichi⍟**

Ophthalmology and Visual Science Department, Hiroshima University, Hiroshima, Japan

\* nakao@hiroshima-u.ac.jp

**Data Availability Statement:** All our study datas are available from the figshare database (10.6084/m9.figshare.11954748).

## Abstract

### Purpose

The aim of the study was to investigate the effects of various anatomical structures on intraocular pressure (IOP) measurements obtained by the Corneal Visualization Scheimpflug Technology (Corvis ST), Goldmann applanation tonometer (GAT), and noncontact tonometer (NCT), as well as to assess the interchangeability among the four types of IOP measurement: IOP-GAT, IOP-NCT, IOP-Corvis, and biomechanically corrected IOP (bIOP-Corvis), with a particular focus on bIOP-Corvis.

### Materials and methods

We included 71 patients with primary open-angle glaucoma and assessed their IOP measurements obtained with the GAT, NCT, and Corvis ST using a repeated measures ANOVA, a paired t-test with Bonferroni correction, stepwise multiple regression analyses and Bland–Altman plots.

### Results

IOP-GAT showed the highest values (13.5 ± 2.1 mmHg [mean ± standard deviation]), followed by IOP-NCT (13.2 ± 2.7 mmHg), IOP-Corvis (10.6 ± 2.8 mmHg), and bIOP-Corvis (10.0 ± 2.3 mmHg). With exceptions of bIOP-Corvis and IOP-GAT, all IOP variations were explained by regression coefficients involving the central corneal thickness. Bland–Altman plots showed a mean difference between IOP-GAT and the other IOP measurements (IOP-Corvis, bIOP-Corvis, and IOP-NCT), which were -2.90, -3.48, and -0.29 mmHg, respectively. The widths of the 95% limits of agreement between all pairs of IOP measurements were greater than 3 mmHg.

**Funding:** The author(s) received no specific funding for this work.

**Competing interests:** The authors have declared that no competing interests exist.

## Conclusion

IOP values obtained with the Corvis ST, NCT, and GAT were not interchangeable. The bIOP-Corvis measurement corrected for the ocular structure.

## Introduction

Intraocular pressure (IOP) is the only treatable risk factor in the management of patients with glaucoma. Previous studies have shown that even a mean IOP increase of 1 mmHg may substantially increase the risk for development and progression of glaucoma [1]. Therefore, precise and accurate assessment of IOP is crucial for proper management of patients with glaucoma. An ideal tonometer should be precise, accurate, and minimally influenced by factors, such as ocular biomechanical parameters. The Goldmann applanation tonometer (GAT) is presently the gold standard for clinical IOP measurements, and the traditional noncontact tonometer (NCT) is widely used in optometric practice as it is rapid and simple to operate with respect to measurement of IOP. Nevertheless, IOP readings increased as the CCT increased [2–12], axial length decreased [13, 14], age increased [9, 12], and the corneal curvature decreased [8, 15] when the two tonometers are used.

In 2005, to provide a corrected IOP unaffected by CCT, the Ocular Response Analyzer (ORA; Reichert, Delpew, NY, USA) was introduced [16]. The ORA was the first non-contact tonometer to convert corneal biomechanics into numerical values using dynamic infrared signal analysis of the corneal biomechanical response. However, the ORA cannot provide a direct description of the mechanical behavior of the cornea.

More recently, the Corneal Visualization Scheimpflug Technology (Corvis ST; Oculus, Wetzlar, Germany) has been introduced as a novel non-contact tonometer designed to accurately measure IOP and the detailed biomechanical response of the cornea to an air pulse. The Corvis ST records the corneal reaction to a defined air pulse with a Scheimpflug imaging system that takes about 4,330 images per second. Then, it estimates the corrected IOP without the influence of ocular biomechanical parameters, including CCT or aging; it is named as IOPpachy-Corvis. Previously, we compared the IOPpachy-Corvis and the IOP obtained with GAT; it is named as IOP-GAT, and found that the two measurements were not interchangeable. This may be because IOPpachy-Corvis was not sufficiently unaffected by the ocular biomechanical parameters [11]. However, in 2016, Corvis ST developed a new parameter called biomechanically corrected IOP (bIOP-Corvis). This new parameter accounts for the dynamic corneal response in addition to the anatomical corneal structures. It is an estimate of the corrected IOP that is minimally influenced by the ocular biomechanical parameters, such as age, CCT, and radius at the highest concavity. The bIOP-Corvis formula [17, 18] is as follows:

bIOP = $C_{CCT1} \times C_{AP1} \times C_{age1} + C_{CCT2} \times C_{age2} + C_{DCR} + a_{19}$

where

$C_{CCT1} = (a_1 \times CCT^3 + a_2 \times CCT^2 + a_3 \times CCT + a_4)$

$C_{AP1} = (a_5 \times AP1 + a_6)$

$C_{age1} = (a_7 \times [Ln(Beta)]^2 + a_8 \times [Ln(Beta)] + a_9)$

$C_{CCT2} = (a_{10} \times CCT^3 + a_{11} \times CCT^2 + a_{12} \times CCT + a_{13})$

$C_{age2} = (a_{14} \times [Ln(Beta)]^2 + a_{15} \times [Ln(Beta)] + a_{16})$

Beta = $0.5852 \times exp(0.0111 \times age[year])$

$C_{DCR} = a_{17} \times$ highest concavity radius $+ a_{18}$

Thus, the biomechanical parameter correction performed is more precise.

The aim of this study was to investigate the effects of various anatomical structures on the IOP measurements obtained with the GAT, NCT, and Corvis ST as well as to assess the interchangeability among four types of IOP measurements, IOP-GAT, IOP-NCT, IOP-Corvis, and bIOP-Corvis, with a particular focus on bIOP-Corvis.

## Materials and methods

This prospective and comparative analysis of IOP values was performed in Hiroshima University Hospital on 71 right eyes of 71 participants with primary open-angle glaucoma (POAG) recruited from September 2014 to March 2015. The institutional board of Hiroshima University Hospital approved the study and waived the need for informed consent owing to the prospective chart review that was created for explanation of the implications of such activities and listed on a poster in the hospital. This study was registered with the University Hospital Medical Network clinical trials registry, and the registration number was JPRN-U-MIN000016623. The authors adhered to the tenets of the Declaration of Helsinki.

The exclusion criteria included intraocular surgery or refractive laser treatment and any systemic or ocular pathology that could affect the IOP measurements; refractive error equal to or exceeding -6.00 diopter equivalent sphere, and corneal astigmatism equal to or exceeding 3.00 diopters; ocular hypertension; diabetes [19]; pregnancy [20]; and Scheimpflug images with a low quality, which cannot be automatically analyzed.

All patients underwent the following examinations on the same day: complete ophthalmologic examination, including spherical equivalent refraction and the average of the horizontal and vertical corneal curvatures (KR-800 ®; Topcon Corporation, Tokyo, Japan, axial length (IOL master ®; Carl Zeiss Meditec AG, Jena, Germany), as well as CCT (Corvis ST). IOP values were obtained with the Corvis ST, CT-90A (Topcon Corporation, Tokyo, Japan) and GAT. Experienced clinicians measured the IOP thrice using each device between 10:00 and 17:00. First, in all cases, the IOP measurements were obtained in a randomized order–CT-90A or Corvis ST–by the same clinicians with a 5-min interval between use of each device. After another 30-min interval, topical anesthesia with 0.4% oxybuprocaine hydrochloride and fluorescein staining was induced, following which, GAT measurements were taken by a masked ophthalmologist.

## Statistical analysis

IOP measurements were compared using a repeated measures ANOVA and a paired t-test with Bonferroni correction. A sample size of 50 participants was needed to achieve 80% power to detect a minimal clinically important difference of 1.5 mmHg, assuming a standard deviation of 1.75 mmHg in the differences between IOPs, a significance level of 0.05, and a 10% patient ineligibility. We utilized univariate regression models to study factors (age, corneal curvature, axial length, and CCT) associated with each IOP measurement obtained with three tonometers. Subsequently, all-subsets and stepwise multivariate linear regression analyses were utilized to construct models that best identified the independent factors associated with the IOP measurements. We used the variance inflation factor (VIF) for each term in the mode used for the potential collinearity problem. A VIF was equal to or exceeding 5.0 indicated a collinearity issue among the terms in the multivariate regression analyses. The target sample size estimates were based on the effect size $f^2$ (0.15); number of predictors (4); significance level (0.05); and power (80%). Considering 10% of patients to be ineligible, the target sample size was determined to be at least 61 patients. The 95% limits of agreement (LOAs) between methods (the mean difference ± 1.96 SD contained 95% of the inter-method differences) were

evaluated using Bland–Altman plots, which also assessed simultaneous visual examinations for both fixed and proportional biases. P-values less than 0.05 were considered significant.

## Results

Demographic data are summarized in Table 1.

Corvis ST, NCT, and GAT were used to measure the IOP of the right eyes of 71 participants.

Overall, the IOP-GAT measurements had the highest values ($13.5 \pm 2.1$ mmHg [mean ± SD]), followed by IOP-NCT ($13.2 \pm 2.7$ mmHg), IOP-Corvis ($10.6 \pm 2.8$ mmHg), and bIOP-Corvis ($10.0 \pm 2.3$ mmHg). We found a significant difference between the IOP measurements by using a repeated measures ANOVA (all, $p < 0.001$). In the paired t-test with Bonferroni correction, bIOP-Corvis obtained significantly the lowest values of the four IOP measurements (all, $p < 0.001$). IOP-Corvis produced significantly lower values than IOP-NCT and IOP-GAT (all, $p < 0.001$); however, we found no significant differences between IOP-NCT and IOP-GAT ($p > 0.05$).

Using univariate regression analyses, only CCT was associated with IOP-Corvis and IOP-NCT (Table 2). We used stepwise multivariate regression analyses to adjust for the interactions among variables. CCT independently influenced IOP-Corvis (standardized $\beta = 0.35$; $p = 0.003$) and IOP-NCT (standardized $\beta = 0.42$; $p = 0.0003$). None of these factors influenced bIOP-Corvis or IOP-GAT. The VIFs of identified factors in the stepwise multivariate regression analysis ranged from 1.0 to 1.1.

Fig 1 shows the Bland–Altman plots for the agreement between the IOP measurements.

The mean difference between IOP-Corvis and IOP-GAT was -2.90 mmHg; the 95% LOA was 3.80 mmHg, and a fixed bias was present ($p < 0.0001$), but we detected a weak proportional bias ($r^2 = 0.15$; $p = 0.0008$) (Fig 1A). The results of a comparison between bIOP-Corvis and IOP-GAT show that mean difference was -3.48 mmHg, and the 95% LOA was the narrowest at 3.42 mmHg. We identified a fixed bias ($p < 0.0001$) but did not detect any proportional bias ($r^2 = 0.02$; $p = 0.21$) (Fig 1B). In comparison between IOP-NCT and IOP-GAT, the mean difference was -0.29 mmHg, the 95% LOA was 3.72 mmHg, and no fixed bias was present ($p = 0.20$). However, we identified a weak proportional bias ($r^2 = 0.14$; $p = 0.0016$) (Fig 1C).

## Discussion

This study investigated the effects of various anatomical structures on IOP measurements obtained with three devices as well as the interchangeability among four types of IOP measurements. The findings in this study indicated that no anatomical factors were associated with bIOP-Corvis or IOP-GAT. Furthermore, the comparison between bIOP-Corvis and IOP-GAT resulted in the narrowest 95% LOA (3.42 mmHg) and no proportional bias. A fixed bias of the

**Table 1. Demographics and ocular characteristics of patients (POAG, n = 71).**

|  | Mean ± SD | Range |
|---|---|---|
| Visual acuity (logMAR) | -0.005 ± 0.21 | -1.08–1.00 |
| Age (year) | 62.75 ± 11.68 | 30.8–79.4 |
| Sex (n, female) | 31 | |
| Axial length (mm) | 25.16 ± 1.74 | 22.1–29.2 |
| Average corneal curvature (mm) | 7.73 ± 0.28 | 7.14–8.45 |
| Central corneal thickness (μm) | 536 ± 33.85 | 460–635 |

POAG, primary open-angle glaucoma; logMAR, logarithm of the minimum angle of resolution; SD, standard deviation

**Table 2. Factors independently associated with IOP measurements in the univariate and multiple regression analyses.**

| Independent variables | IOP measurements | | | | | | | | | | | |
|---|---|---|---|---|---|---|---|---|---|---|---|---|
| | bIOP-Corvis | | | IOP-Corvis | | | IOP-NCT | | | IOP-GAT | | |
| | Standardized β | p | VFI | Standardized β | p | VFI | Standardized β | p | VFI | Standardized β | p | VFI |
| Univariate regression analysis | | | | | | | | | | | | |
| Age (year) | -0.13 | 0.282 | | 0.00 | 0.992 | | -0.161 | 0.180 | | 0.004 | 0.976 | |
| Average corneal curvature (mm) | -0.16 | 0.177 | | -0.18 | 0.142 | | 0.076 | 0.527 | | -0.050 | 0.677 | |
| Axial length (mm) | 0.05 | 0.697 | | 0.04 | 0.748 | | 0.164 | 0.172 | | 0.030 | 0.802 | |
| Central corneal thickness (μm) | 0.01 | 0.926 | | 0.34 | 0.004 | | 0.419 | 0.000 | | 0.179 | 0.136 | |
| Stepwise multivariate regression analysis | | | | | | | | | | | | |
| Age (year) | | | 1.1 | | | | | | | | | |
| Average corneal curvature (mm) | | | 1.1 | -0.19 | 0.089 | 1.0 | | | | | | |
| Axial length (mm) | | | | | | | | | | | | |
| Central corneal thickness (μm) | | | | 0.35 | 0.003 | 1.0 | 0.419 | 0.000 | | | | |

VFI, variance inflation factor; IOP, intraocular pressure; IOP-Corvis indicates IOP by Corvis ST; bIOP-Corvis, corrected IOP-Corvis; IOP-GAT, IOP by Goldmann applanation tonometry; IOP-NCT, the IOP obtained by CT-90A tonometer.

comparison between bIOP-Corvis and GAT-IOP showed the highest value (-3.48 mmHg); therefore, bIOP-Corvis significantly underestimated IOP-GAT.

## Factors affecting the corrected IOP and noncorrected IOP measurements

Many factors can influence the measurement accuracy. Factors that influence IOP measurements are mostly CCT [2–7, 9, 10, 21], corneal curvature [15], and axial length [13, 14]. In this study, the stepwise multivariate linear regression analyses were used to detect the anatomical and structural factors associated with IOP measurements. CCT was the only significant factor, and it was associated with IOP-Corvis (standardized β = 0.35; $p$ = 0.003) and IOP-NCT (standardized β = 0.42; $p$ = 0.0003) but not with bIOP-Corvis or IOP-GAT (standardized β = 0.179; $p$ = 0.136). The bIOP-Corvis values were calibrated to eliminate the effect of CCT but the IOP-Corvis values were not. Moreover, GAT and NCT are generally affected by CCT, but NCT is more influenced by CCT than GAT [3, 5–7]. This outcome may be explained by these reasons.

## Agreement between the three IOP measurements and IOP-GAT

To date, many studies have assessed the agreement among bIOP-Corvis, IOP-Corvis, IOP-GAT, and IOP-NCT. There was a fixed bias in the comparison between the three IOP readings

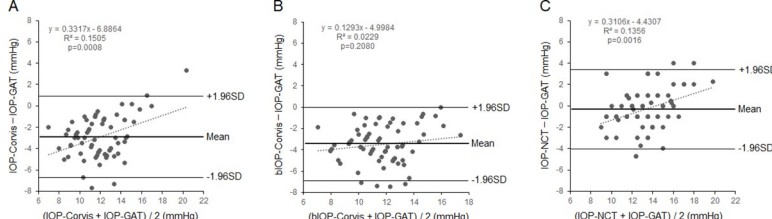

**Fig 1. Bland–Altman plots between IOPs obtained with the Corvis ST, GAT, and CT-90A.** A, IOP-Corvis and IOP-GAT. B, bIOP-Corvis and IOP-GAT. C, IOP-NCT and IOP-GAT. The mean values and 95% LOA are indicated by bold lines and solid lines, respectively. IOP, intraocular pressure; IOP-Corvis, the IOP obtained by Corvis ST; bIOP-Corvis, corrected IOP-Corvis; IOP-GAT, the IOP obtained by Goldmann applanation tonometry; IOP-NCT, the IOP obtained by CT-90A tonometer; LOA, limits of agreement.

and the IOP-GAT values. Bland–Altman plots revealed that the fixed bias of the comparison between bIOP-Corvis and IOP-GAT values, IOP-Corvis and IOP-GAT values, and IOP-NCT and IOP-GAT values in the present study were -3.48, -2.90, -0.29 mmHg, respectively. Recently, Vinciguerra R et al. [22] reported the mean differences between bIOP-Corvis and IOP-GAT. In healthy control eyes and eyes with ocular hypertension, POAG, and normal-tension glaucoma, IOP-GAT were 16.4 ± 2.4, 22.1 ± 4.8, 17.2 ± 4.9, and 13.7 ± 1.8, respectively, and bIOP-Corvis were 13.4 ± 2.8, 17.0 ± 4.1, 14.8 ± 3.1, and 12.9 ± 2.3, respectively. In other words, the bIOP-Corvis values were smaller than the GAT-IOP values in all groups. In a previous study, the fixed bias [10, 21, 23–25] of the comparison between IOP-Corvis and IOP-GAT and the fixed bias [23, 26–28] of the comparison between IOP-NCT and IOP-GAT were smaller than the fixed bias of the comparison between bIOP-Corvis and IOP-GAT.

In our study, no proportional bias was present for comparisons between bIOP-Corvis and IOP-GAT; however, a proportional bias was present for comparisons between IOP-Corvis and IOP-GAT and between IOP-NCT and IOP-GAT. In a previous study, the proportional bias was identically present for IOP-Corvis and IOP-GAT [10, 21]. This showed that the differences between two IOP values (bIOP-Corvis and IOP-GAT) neither increase nor decrease in proportion to the average values; therefore, it can be easily converted from bIOP-Corvis to IOP-GAT.

In our study, the relationships between the 95% LOAs were as follows: The bIOP-Corvis and IOP-GAT had the lowest of the three 95% LOA widths of 3.42 mmHg, and the 95% LOAs between IOP-Corvis and IOP-GAT and between IOP-NCT and IOP-GAT were 3.80 and 3.72 mmHg, respectively. Most recently, Ye Y et al. showed that the 95% LOAs between bIOP-Corvis and IOP-GAT was 3.84 mmHg [12]. In many studies, the 95% LOAs between IOP-Corvis and IOP-GAT ranged widely: +4.40 [24], +4.80 [23], +5.40 [10], +6.05 [21], and +8.00 [25] mmHg. In some studies, the 95% LOAs between IOP-NCT and IOP-GAT also had a wide range: +2.17 [26], +3.30 [27], +4.60 [28], and +7.20 [23] mmHg, which means that bIOP-Corvis steadily measures IOP readings accurately when compared to the non-corrected IOP measurements in IOP-Corvis and IOP-NCT.

## Advantages and disadvantages of bIOP-Corvis

Most commercially available tonometers estimate IOP based on the corneal applanation or indentation. Therefore, measured values of IOP are generally influenced by corneal biomechanics. The bIOP-Corvis is an estimate of the corrected IOP, which is minimally influenced by ocular biomechanical parameters. In this study, we used regression analyses to investigate the anatomical and structural factors that affect the bIOP-Corvis measurements. We found that age, average corneal curvature, axial length, and CCT were not significant factors that influence bIOP-Corvis. However, our result of the Bland–Altman plots for bIOP-Corvis and IOP-GAT showed that fixed biases were identified. This means that bIOP-Corvis significantly underestimated IOP using the standard clinical tonometer for measurements. Nevertheless, proportional bias was not between the bIOP-Corvis and IOP-GAT measurements and the 95% LOAs was relatively low. Therefore, it seems that we can convert from bIOP-Corvis to IOP-GAT. Our result implied that the bIOP-Corvis is a useful IOP value in patients with POAG.

Our study has several limitations. First, it is still not clear which IOP measurement is closest to the true value. We evaluated the Corvis ST and CT-90A in comparison with GAT; however, the IOP-GAT is not a true IOP value. Second, our participants included only patients with POAG. It is unclear whether our results can be applicable to healthy patients and those without POAG. Third, our participants were treated using antiglaucoma medications, which are

intended to reduce elevated intraocular pressure. Thus, the IOPs measured in our participants all seemed within the normal range. Our results may not generalize to those with higher IOP measurements. Fourth, our study did not consider other factors that may influence IOP readings, such as visual acuity [29, 30] and number of antiglaucoma medications [31, 32], and future research is desired in this aspect.

In conclusion, we showed that no anatomical factors were associated with bIOP-Corvis. The comparison between bIOP-Corvis and IOP-GAT resulted in the narrowest 95% LOA with no proportional bias. We can convert bIOP-Corvis to IOP-GAT by correcting the fixed biases. The Corvis ST devices offer new possibilities for clinically useful tonometers.

## Acknowledgments

We would like to thank Editage (www.editage.com) for English language editing.

## Author Contributions

**Conceptualization:** Yoshiaki Kiuchi.

**Formal analysis:** Yoshitaka Nakao.

**Funding acquisition:** Yoshiaki Kiuchi.

**Investigation:** Yoshitaka Nakao, Hideaki Okumichi.

**Methodology:** Yoshitaka Nakao, Yoshiaki Kiuchi.

**Project administration:** Yoshitaka Nakao, Yoshiaki Kiuchi.

**Software:** Yoshiaki Kiuchi.

**Supervision:** Yoshiaki Kiuchi.

**Visualization:** Yoshitaka Nakao.

**Writing – original draft:** Yoshitaka Nakao.

**Writing – review & editing:** Yoshiaki Kiuchi, Hideaki Okumichi.

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
