## [Decision Letter · Decision Letter 0]

5 May 2020

PONE-D-20-07468

Evaluation of biomechanically corrected intraocular pressure using Corvis ST and comparison of the Corvis ST, noncontact tonometer, and Goldmann applanation tonometer in patients with glaucoma

PLOS ONE

Dear %Dr% %Nakao%,

Thank you for submitting your manuscript to PLOS ONE. After careful consideration, we feel that it has merit but does not fully meet PLOS ONE’s publication criteria as it currently stands. Therefore, we invite you to submit a revised version of the manuscript that addresses the points raised during the review process.

We would appreciate receiving your revised manuscript by Jun 19 2020 11:59PM. To enhance the reproducibility of your results, we recommend that if applicable you deposit your laboratory protocols in protocols.io, where a protocol can be assigned its own identifier (DOI) such that it can be cited independently in the future. For instructions see: http://journals.plos.org/plosone/s/submission-guidelines#loc-laboratory-protocols

We look forward to receiving your revised manuscript.

Kind regards,

Rajiv R. Mohan, Ph.D.

Academic Editor

PLOS ONE

Additional Editor Comments (if provided):

Dear authors,

I wish to inform that your MSS has been reviewed by the two reviewers and EBM, and requires addressing of some key points/issues. It is recommended to read the concerns and address them appropriately for the consideration of publication.

Thank you

Rajiv Mohan

Reviewers' comments:

Reviewer's Responses to Questions

**Comments to the Author**

1. Is the manuscript technically sound, and do the data support the conclusions?

Reviewer #1: Partly

Reviewer #2: Yes

2. Has the statistical analysis been performed appropriately and rigorously? 

Reviewer #1: Yes

Reviewer #2: No

3. Have the authors made all data underlying the findings in their manuscript fully available?

Reviewer #1: Yes

Reviewer #2: Yes

4. Is the manuscript presented in an intelligible fashion and written in standard English?

Reviewer #1: Yes

Reviewer #2: Yes

5. Review Comments to the Author

Reviewer #1: This is an interesting and current article but needs adjustments such as improving the explanation in the introduction on the thickness and biomechanics of the cornea, citing the competing device (ORA) that was the first to use this type of technology and giving more details on how Corvis works ST.

In the methodology, you should mention inclusion criteria (minimum visual acuity, how many drugs the patient is using, minimum and maximum IOP included in the work), think about including pregnant women and ocular hypertensive patients in the exclusion criteria and better explain the power of the sample.

It uses adequate statistics and correlates well the data of the proposed methodology.

In the discussion, he concludes about the proposed methodology and results and reports the negative points. Among them, there is a fundamental issue that would be the inclusion of a control group to correlate and compare data from patients with glaucoma.

Reviewer #2: The paper is written to investigate the IOP measured by Corvis, NCT and GAT. However, there are still some questions existed and need to be checked.

Major comments

1. The IOP values in glaucoma is not exactly same to normal eyes, it should add the comparsion of the IOP measured by Corvis, NCT and GAT in normal eyes.

2. The author found the IOP-GAT measurements had the highest values (13.5 ± 2.1 mmHg [mean ± SD]), followed by IOP-NCT (13.2 ± 2.7 mmHg), IOP-Corvis (10.6 ± 2.8 mmHg), and bIOP-Corvis (10.0 ± 2.3 mmHg). The statistically analysis (such as analysis of variance, and SNK method) is needed to compare the IOP values.

3. In Table 2, the author used the univariate and multivariate regression analyses. There are no significantly differences of bIOP-Corvis and IOP-GAT in the univariate results, which should not further conduct multivariate regression analyses, please checks it.

4. The clinical application of four IOP values needs to be discussed and the discussion should more deeply.

Minor comments.

1. Page 14 line 12, check the sentence “This showed that that the……”

2. the Corvis ST only occurred in title, which is not mentioned in main documents.

6. PLOS authors have the option to publish the peer review history of their article (what does this mean?). If published, this will include your full peer review and any attached files.

Reviewer #1: No

Reviewer #2: No

---

## [Author Response · Author response to Decision Letter 0]

27 May 2020

Dear Dr Rajiv R. Mohan, we would like to thank you and the reviewers for their constructive and insightful reviews. We believe that the points raised, and our revisions, have resulted in an improved manuscript again. Our responses to the suggested revisions follow.

Reviewer #1: 

Thank you very much. Please find the comments and corrections added in the manuscript.

This is an interesting and current article but needs adjustments such as improving the explanation in the introduction on the thickness and biomechanics of the cornea, citing the competing device (ORA) that was the first to use this type of technology and giving more details on how Corvis works ST.

Thank you very much for this comment. We agree with you. We added the sentence below in the introduction section (Page ４ Line 52).

“Nevertheless, IOP readings increased as the CCT increased [2-12], axial length decreased [13, 14], age increased [9, 12], and the corneal curvature decreased [8, 15] when the two tonometers are used.

 In 2005, to provide a corrected IOP unaffected by CCT, the Ocular Response Analyzer (ORA; Reichert, Delpew, NY, USA) was introduced [16]. The ORA was the first non-contact tonometer to convert corneal biomechanics into numerical values using dynamic infrared signal analysis of the corneal biomechanical response. However, the ORA cannot provide a direct description of the mechanical behavior of the cornea.

 More recently, the Corneal Visualization Scheimpflug Technology (Corvis ST; Oculus, Wetzlar, Germany) has been introduced as a novel non-contact tonometer designed to accurately measure IOP and the detailed biomechanical response of the cornea to an air pulse. The Corvis ST records the corneal reaction to a defined air pulse with a Scheimpflug imaging system that takes about 4,330 images per second. Then, it estimates the corrected IOP without the influence of ocular biomechanical parameters, including CCT or aging; it is named as IOPpachy-Corvis.”

In the methodology, you should mention inclusion criteria (minimum visual acuity, how many drugs the patient is using, minimum and maximum IOP included in the work), think about including pregnant women and ocular hypertensive patients in the exclusion criteria and better explain the power of the sample.

Thank you very much. We totally agree with you. We did not consider the inclusion criteria about minimum visual acuity, how many drugs the patient is using, minimum and maximum IOP in advance. As discussed above, we added the limitations in the discussion section (Page 18 Line 266). 

 “Second, our participants included only patients with POAG. It is unclear whether our results can be applicable to healthy patients and those without POAG. Third, our participants were treated using antiglaucoma medications, which are intended to reduce elevated intraocular pressure. Thus, the IOPs measured in our participants all seemed within the normal range. Our results may not generalize to those with higher IOP measurements. Fourth, our study did not consider other factors that may influence IOP readings, such as visual acuity [29, 30] and number of antiglaucoma medications [31, 32], and future research is desired in this aspect.”

And we added the visual acuity data (mean±standard deviation and range) in the table1 (Page 9 Line 142)

Table 1. Demographics and ocular characteristics of patients (POAG, n=71)

We include a pregnant women and ocular hypertensive patients in the exclusion criteria in the Materials and Methods section (Page7 Line107). 

“The exclusion criteria included intraocular surgery or refractive laser treatment and any systemic or ocular pathology that could affect the IOP measurements; refractive error equal to or exceeding -6.00 diopter equivalent sphere, and corneal astigmatism equal to or exceeding 3.00 diopters; ocular hypertension; diabetes [19]; pregnancy [20]; and Scheimpflug images with a low quality, which cannot be automatically analyzed.”

And we tried to explain the power of the sample in the statistical analysis section (Page 8 Line 122/ Page 9 Line 132).

“IOP measurements were compared using a repeated measures ANOVA and a paired t-test with Bonferroni correction. A sample size of 50 participants was needed to achieve 80% power to detect a minimal clinically important difference of 1.5 mmHg, assuming a standard deviation of 1.75 mmHg in the differences between IOPs, a significance level of 0.05, and a 10% patient ineligibility.”

“The target sample size estimates were based on the effect size f2 (0.15); number of predictors (4); significance level (0.05); and power (80%). Considering 10% of patients to be ineligible, the target sample size was determined to be at least 61 patients. "

It uses adequate statistics and correlates well the data of the proposed methodology.

In the discussion, he concludes about the proposed methodology and results and reports the negative points. Among them, there is a fundamental issue that would be the inclusion of a control group to correlate and compare data from patients with glaucoma.

Thank you so much. We totally agree with you. In the future, we need to compare between healthy subjects and patients with glaucoma. We added the sentence below to the limitation in the discussion section (Page 18 Line 266).

“Second, our participants included only patients with POAG. It is unclear whether our results can be applicable to healthy patients and those without POAG.”

Reviewer #2: The paper is written to investigate the IOP measured by Corvis, NCT and GAT. However, there are still some questions existed and need to be checked.

Thank you so much. Please find the comments and corrections added in the manuscript.

Major comments

1. The IOP values in glaucoma is not exactly same to normal eyes, it should add the comparsion of the IOP measured by Corvis, NCT and GAT in normal eyes.

Thank you very much. We totally agree with you and Reviewer1. In the future, we need to compare between healthy subjects and patients with glaucoma. We added the sentence below to the limitation in the discussion section (Page 18 Line 266).

“Second, our participants included only patients with POAG. It is unclear whether our results can be applicable to healthy patients and those without POAG.”

2. The author found the IOP-GAT measurements had the highest values (13.5 ± 2.1 mmHg [mean ± SD]), followed by IOP-NCT (13.2 ± 2.7 mmHg), IOP-Corvis (10.6 ± 2.8 mmHg), and bIOP-Corvis (10.0 ± 2.3 mmHg). The statistically analysis (such as analysis of variance, and SNK method) is needed to compare the IOP values.

Thank you very much. We agree with your opinion. As discussed above, we decided to utilize a repeated measures ANOVA and a paired t-test with Bonferroni correction. 

We added the sentence below in the abstract section (Page 2 Line 26)

“We included 71 patients with primary open-angle glaucoma and assessed their IOP measurements obtained with the GAT, NCT, and Corvis ST using a repeated measures ANOVA, a paired t-test with Bonferroni correction, stepwise multiple regression analyses and Bland–Altman plots.”

And in the statistical analysis section (Page 8 Line 122)

“IOP measurements were compared using a repeated measures ANOVA and a paired t-test with Bonferroni correction.”

And in the results section (Page 10 Line 155).

“We found a significant difference between the IOP measurements by using a repeated measures ANOVA (all, p <0.001). In the paired t-test with Bonferroni correction, bIOP-Corvis obtained significantly the lowest values of the four IOP measurements (all, p <0.001). IOP-Corvis produced significantly lower values than IOP-NCT and IOP-GAT (all, p <0.001); however, we found no significant differences between IOP-NCT and IOP-GAT (p >0.05).”

3. In Table 2, the author used the univariate and multivariate regression analyses. There are no significantly differences of bIOP-Corvis and IOP-GAT in the univariate results, which should not further conduct multivariate regression analyses, please checks it.

Thank you so much. We agree with you. We do not need to further conduct multivariate regression analyses for bIOP-Corvis and IOP-GAT. We corrected the table 2 (Page 12)

4. The clinical application of four IOP values needs to be discussed and the discussion should more deeply.

Thank you very much. As discussed above, we added the sentence below in the discussion section (Page 17 Line 251).

“Advantages and disadvantages of bIOP-Corvis

 Most commercially available tonometers estimate IOP based on the corneal applanation or indentation. Therefore, measured values of IOP are generally influenced by corneal biomechanics. The bIOP-Corvis is an estimate of the corrected IOP, which is minimally influenced by ocular biomechanical parameters. In this study, we used regression analyses to investigate the anatomical and structural factors that affect the bIOP-Corvis measurements. We found that age, average corneal curvature, axial length, and CCT were not significant factors that influence bIOP-Corvis. However, our result of the Bland–Altman plots for bIOP-Corvis and IOP-GAT showed that fixed biases were identified. This means that bIOP-Corvis significantly underestimated IOP using the standard clinical tonometer for measurements. Nevertheless, proportional bias was not between the bIOP-Corvis and IOP-GAT measurements and the 95% LOAs was relatively low. Therefore, it seems that we can convert from bIOP-Corvis to IOP-GAT. Our result implied that the bIOP-Corvis is a useful IOP value in patients with POAG.”

Minor comments.

1. Page 14 line 12, check the sentence “This showed that that the……”

Thank you so much. We corrected the sentence (Page 16 Line 236).

“This showed that the differences between two IOP values (bIOP-Corvis and IOP-GAT) neither increase nor decrease in proportion to the average values; therefore, it can be easily converted from bIOP-Corvis to IOP-GAT.”

2. the Corvis ST only occurred in title, which is not mentioned in main documents.

Thank you very much. We added the sentence below in the abstract section (Page 2 Line 19)

“The aim of the study was to investigate the effects of various anatomical structures on intraocular pressure (IOP) measurements obtained by the Corneal Visualization Scheimpflug Technology (Corvis ST)”

And in the introduction section (Page 5 Line 64).

“More recently, the Corneal Visualization Scheimpflug Technology (Corvis ST; Oculus, Wetzlar, Germany) has been introduced as a novel non-contact tonometer designed to accurately measure IOP and the detailed biomechanical response of the cornea to an air pulse. The Corvis ST records the corneal reaction to a defined air pulse with a Scheimpflug imaging system that takes about 4,330 images per second.”

And we added the words ”Corvis ST” below in Page 5 Line73/ Page6 Line 91/ Page 7 Line 112/ Page 8 Line 114, 117/ Page 10 Line 152/ Page 13 Line 179, 182/ Page 17 Line 249/ Page 18 Line 265/ Page 19 Line 285

---

## [Decision Letter · Decision Letter 1]

17 Aug 2020

Evaluation of biomechanically corrected intraocular pressure using Corvis ST and comparison of the Corvis ST, noncontact tonometer, and Goldmann applanation tonometer in patients with glaucoma

PONE-D-20-07468R1

Dear Dr. Nakao,

We’re pleased to inform you that your manuscript has been judged scientifically suitable for publication and will be formally accepted for publication once it meets all outstanding technical requirements.

Kind regards,

Rajiv R. Mohan, Ph.D.

Academic Editor

PLOS ONE

Additional Editor Comments (optional):

Reviewers' comments:

Reviewer's Responses to Questions

**Comments to the Author**

1. If the authors have adequately addressed your comments raised in a previous round of review and you feel that this manuscript is now acceptable for publication, you may indicate that here to bypass the “Comments to the Author” section, enter your conflict of interest statement in the “Confidential to Editor” section, and submit your "Accept" recommendation.

Reviewer #2: All comments have been addressed

2. Is the manuscript technically sound, and do the data support the conclusions?

Reviewer #2: Yes

3. Has the statistical analysis been performed appropriately and rigorously? 

Reviewer #2: No

4. Have the authors made all data underlying the findings in their manuscript fully available?

Reviewer #2: Yes

5. Is the manuscript presented in an intelligible fashion and written in standard English?

Reviewer #2: Yes

6. Review Comments to the Author

Reviewer #2: The previous question has been revised, and there are still some inaccuracies. The author added ocular hypertension, diabetes and pregnancy to the exclusion criteria, which changed the sample size. Although the results will not change significantly, the authors should perform statistical analysis based on the new sample after exclusion criteria.

7. PLOS authors have the option to publish the peer review history of their article (what does this mean?). If published, this will include your full peer review and any attached files.

Reviewer #2: No

---

## [Editor Report · Acceptance letter]

14 Sep 2020

PONE-D-20-07468R1 

Evaluation of biomechanically corrected intraocular pressure using Corvis ST and comparison of the Corvis ST, noncontact tonometer, and Goldmann applanation tonometer in patients with glaucoma 

Dear Dr. Nakao:

I'm pleased to inform you that your manuscript has been deemed suitable for publication in PLOS ONE. Congratulations! Your manuscript is now with our production department. 

Kind regards, 

on behalf of

Dr. Rajiv R. Mohan 

Academic Editor

PLOS ONE